Resource

# A genetic screen to uncover mechanisms underlying lipid transfer protein function at membrane contact sites

Shirish Mishra[1], Vaishnavi Manohar[1], Shabnam Chandel[1], Tejaswini Manoj[1], Subhodeep Bhattacharya[1], Nidhi Hegde[1], Vaisaly R Nath[1,2], Harini Krishnan[1], Corinne Wendling[3], Thomas Di Mattia[3], Arthur Martinet[3], Prasanth Chimata[1], Fabien Alpy[3], Padinjat Raghu[1]

Lipid transfer proteins mediate the transfer of lipids between organelle membranes, and the loss of function of these proteins has been linked to neurodegeneration. However, the mechanism by which loss of lipid transfer activity leads to neurodegeneration is not understood. In *Drosophila* photoreceptors, depletion of retinal degeneration B (RDGB), a phosphatidylinositol transfer protein, leads to defective phototransduction and retinal degeneration, but the mechanism by which loss of this activity leads to retinal degeneration is not understood. RDGB is localized to membrane contact sites through the interaction of its FFAT motif with the ER integral protein VAP. To identify regulators of RDGB function in vivo, we depleted more than 300 VAP-interacting proteins and identified a set of 52 suppressors of *rdgB*. The molecular identity of these suppressors indicates a role of novel lipids in regulating RDGB function and of transcriptional and ubiquitination processes in mediating retinal degeneration in *rdgB⁹*. The human homologs of several of these molecules have been implicated in neurodevelopmental diseases underscoring the importance of VAP-mediated processes in these disorders.

## Introduction

The maintenance of exact membrane lipid composition is important for providing a distinct identity to cellular organelles and thus for supporting normal cellular physiology (Harayama & Riezman, 2018). Various lipid species reach their specific organelle membrane via either vesicular or non-vesicular transport. Proteins that shuttle lipids in a non-vesicular manner across various compartments are known as lipid transfer proteins (LTPs). Each of these LTPs transfer specific lipid species such as sterols, ceramides, or phospholipids, and in many cases, the LTPs are localized at very specific locations known as membrane contact sites (MCS). In a eukaryotic cell, MCS are regions where two organelle membranes come very close at the range of 10–30 nm but do not fuse (Prinz et al, 2020). Being the largest cellular organelle, the ER forms MCS with the mitochondria, lysosomes, the Golgi network, lipid droplets, and the plasma membrane (PM). MCS provide the fast and efficient delivery of metabolites between two membranes and could be permanent or induced (Wu et al, 2018); this includes the exchange of lipids between organelle membranes to support ongoing cell physiology (Cockcroft & Raghu, 2018). Growing evidence suggests an important role of LTP function at MCS and LTPs in human neurological disorders (Fowler et al, 2019; Peretti et al, 2020; Guillén-Samander & De Camilli, 2022). However, much remains to be discovered on the regulation of LTP function at MCS.

MCS between the ER and the PM are important for regulating both plasma membrane lipid composition and signalling functions. One of the best examples of the requirement of an LTP at the ER-PM MCS is sensory transduction in *Drosophila* photoreceptors (Yadav et al, 2016). Photoreceptors detect light through the G protein–coupled receptor rhodopsin (Rh), leading to the hydrolysis of phosphatidylinositol 4,5-bisphosphate [$PI(4,5)P_2$] by G protein–coupled PLC activity (Hardie & Raghu, 2001). As part of their ecology, fly photoreceptors are exposed to light; in bright daylight, they typically absorb ca. $10^6$ effective photons/second resulting in extremely high PLC activity. Hence, fly photoreceptors provide an excellent model system to study the turnover of $PI(4,5)P_2$ during PLC-mediated cell signalling (Raghu et al, 2012).

Given the low abundance of $PI(4,5)P_2$, replenishment of this lipid at the PM is necessary for uninterrupted PLC signalling. Many enzymes and proteins participate in this process, but a key step is the transfer of lipids that are intermediates of the $PI(4,5)P_2$ cycle. One of the proteins at this site is retinal degeneration B (RDGB), a large multi-domain protein with an N-terminal phosphatidylinositol transfer protein (PITP) domain (Raghu et al, 2021). The PITP domain belongs to the superfamily of LTPs. In the case of RDGB, its PITP domain can transfer phosphatidylinositol (PI) and phosphatidic acid (PA) in vitro (Yadav et al, 2015); a property that is conserved in its mammalian ortholog, Nir2 (Kim et al, 2015). *rdgB* mutant flies undergo light-dependent retinal degeneration, a

[1]National Centre for Biological Sciences-TIFR, GKVK Campus, Bangalore, India  [2]School of Biotechnology, Amrita Vishwa Vidyapeetham, Kollam, India  [3]Université de Strasbourg, CNRS, Inserm, IGBMC UMR 7104- UMR-S 1258, Illkirch, France

Correspondence: praghu@ncbs.res.in

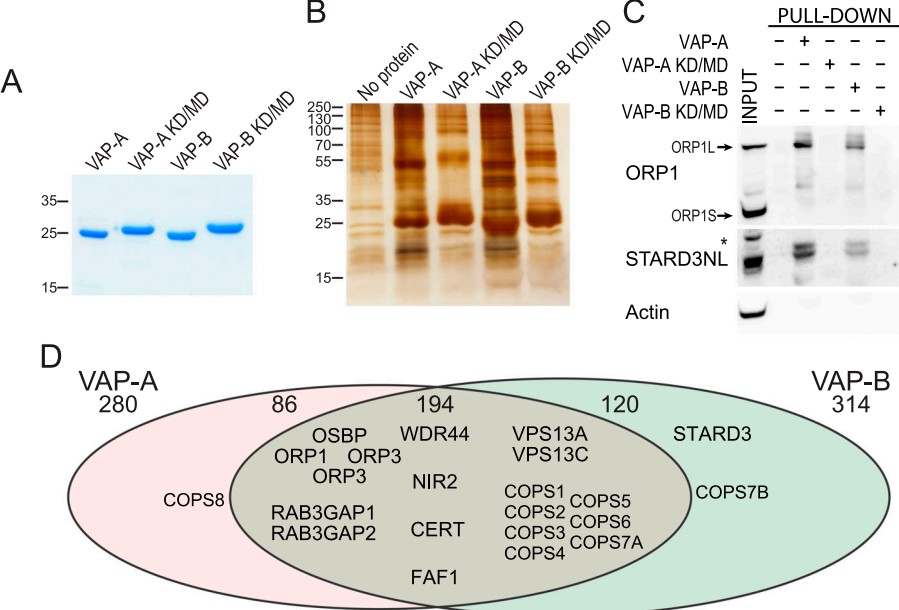

**Figure 1. Identification of VAP-A and VAP-B binding partners.**
**(A)** Coomassie Blue staining of the recombinant WT and KD/MD mutant MSP domains of VAP-A and VAP-B after SDS–PAGE. **(B)** Silver nitrate staining of proteins pulled down using WT MSP domains of VAP-A and VAP-B, and the KD/MD mutant MSP domains, after SDS–PAGE. **(C)** Western blot analysis of proteins pulled down using the WT and mutant MSP domain of VAP-A and VAP-B. The input and pull-down fractions correspond to HeLa cell total protein extract and bound proteins, respectively. *: non-specific band. **(D)** Venn diagram of proteins pulled down by VAP-A and VAP-B (and not by mutant VAP-A and VAP-B). A total of 401 proteins were pulled down with either VAP-A or VAP-B. 194 proteins were pulled down with both VAP-A and VAP-B.

reduced electroretinogram (ERG) response, and a reduced rate of $PI(4,5)P_2$ resynthesis at the PM after PLC activation (Hotta & Benzer, 1970; Harris & Stark, 1977; Yadav et al, 2015). In photoreceptors, RDGB is localized at the ER-PM MCS formed between the microvillar plasma membrane and the SMC, a specialization of the smooth ER (Yadav et al, 2016). The localization of RDGB at this MCS is critically dependent on its interaction with the ER integral membrane protein, VAP (vesicle-associated membrane protein–associated protein). This interaction is physiologically relevant as disruption of the protein–protein interaction between RDGB and VAP in *Drosophila* photoreceptors results in mislocalization of RDGB from this MCS, reduces the efficiency of $PI(4,5)P_2$ turnover, and impacts the response to light (Yadav et al, 2018). However, the mechanisms by which the activity of RDGB is regulated by other proteins at the MCS in this in vivo model system remain to be discovered. VAPs are involved in a range of interactions with proteins containing FFAT/FFNT/Phospho-FFAT/non-FFAT motifs (Slee & Levine, 2019; Cabukusta et al, 2020; Di Mattia et al, 2020). Thus, it seems possible that other proteins involved in regulating biochemical activity at this MCS might also be localized to the sub-microvillar cisternae via VAP interactions. The identification and analysis of proteins engaged in VAP-dependent interactions might help in understanding the regulation of RDGB function. Importantly, VAPs have been implicated in neurodegenerative disorders such as amyotrophic lateral sclerosis, frontotemporal dementia, Alzheimer's disease (AD), and Parkinson's disease (reviewed in Dudás et al [2021]).

In this study, we have carried out a proteomics screen to identify protein interactors of VAP-A and VAP-B in mammalian cells and tested their functional significance in the context of neurodegeneration using the experimental paradigm of RDGB function in *Drosophila* photoreceptors in vivo. The candidates so identified perform a wide range of sub-cellular functions indicating an extensive network of biochemical processes that control the function

of RDGB in regulating lipid transfer during PLC signalling, thus maintaining the structural and functional integrity of neurons.

# Results

## Strategy of proteomics screen

To obtain a list of proteins interacting with VAPs, we performed pull-down experiments in human cells. We produced, in *Escherichia coli*, and purified the MSP domain of human VAP-A and VAP-B fused to a C-terminal 6His-tag (Fig 1A). As a negative control, we used the K94D/M96D and K87D/M89D mutants (herein named KD/MD mutants) of VAP-A and VAP-B, respectively, that are unable to bind FFAT (two phenylalanines in an acidic tract) motifs (Kaiser et al, 2005; Wilhelm et al, 2017). Recombinant proteins were attached to a $Ni^{2+}$-NTA resin and then incubated with protein extracts from HeLa cells. Bound proteins were eluted and analysed by SDS–PAGE followed by silver nitrate staining (Fig 1B) that showed numerous differential bands between WT and mutant VAP samples, suggesting that many proteins are pulled down owing to VAP's ability to bind FFAT motifs. To verify the pull-down efficiency, we performed Western blot using antibodies against two known VAP partners, ORP1 and STARD3NL (Fig 1C) (Rocha et al, 2009; Alpy et al, 2013). ORP1 exists as a long and a short isoform called ORP1L and ORP1S, respectively, ORP1L being the only one of the two to possess an FFAT motif. As expected, the ORP1L isoform was pulled down by WT VAPs but not by mutant VAPs, and the ORP1S isoform was not precipitated (Fig 1C). Besides, STARD3NL co-precipitated with WT VAP-A and VAP-B and not with mutant VAPs, whereas actin, used as a loading control, was not found in the eluted fractions (Fig 1C and Table S1). To identify the proteins pulled down by VAPs, eluates were analysed by tandem

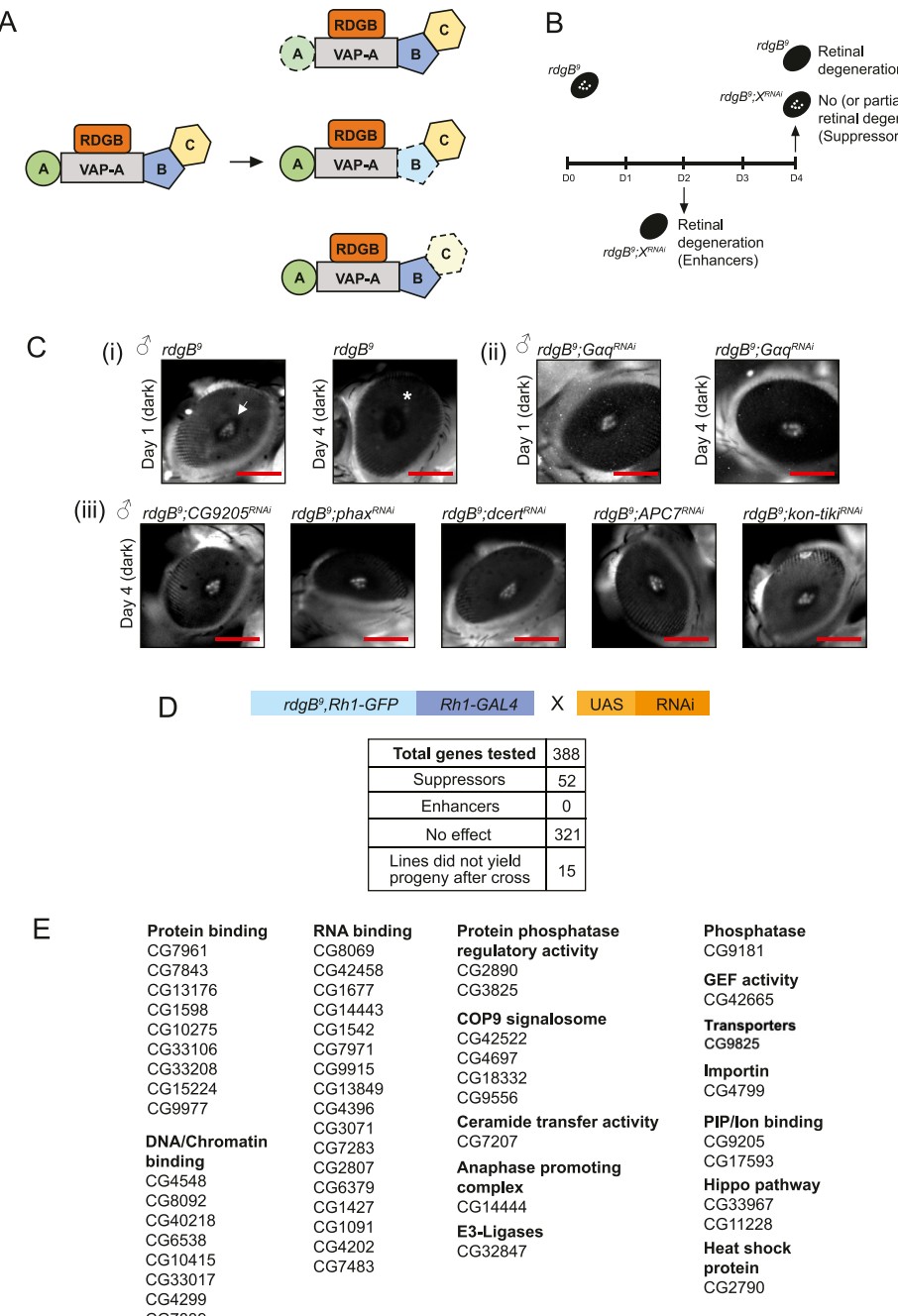

**Figure 2. Strategy of the genetic screen and hits found.**
**(A)** Cartoon depicting classes of VAP interactors used in the present genetic screen. Three classes of genetic interactors of *rdgB* are shown based on the likely molecular mechanism: loss of A, a direct physical interactor of VAP-A; loss of B, a direct interactor of VAP-A that also interacts with C, a protein required for RDGB function; and loss of C, a protein required for *rdgB* function but only interacts with VAP-A via B. Depletion of a specific VAP interactor is depicted with a dotted line. Fly homologs were filtered using DIOPT in FlyBase (http://flybase.org/). **(B)** Genetic scheme used to find either enhancers or suppressors of the retinal degeneration phenotype of *rdgB⁹*. **(C)** Pseudopupil imaging: (i) *rdgB⁹* showed retinal degeneration by day 4 in dark when checked via deep pseudopupil imaging (depicted by *). (ii) The degeneration was partially suppressed when levels of G$_{αq}$ were down-regulated in *rdgB⁹* on day 4. (iii) Selected hits that showed suppression of retinal degeneration in *rdgB⁹* on day 4 (scale bar 225 μm). **(D)** Table showing the full list of genes used in the screen and the number of suppressor genes identified. **(E)** Positive hits (suppressor genes) are divided into different categories depending on their cellular functions. n = 5 flies/RNAi line.

mass spectrometry (MS/MS) (referred to below as IP-MS). The full list of interactions identified by MS/MS in this experiment is shown in Table S2. To identify proteins pulled down according to their ability to interact with VAPs in an FFAT-dependent manner, proteins were ranked based on their enrichment in the WT over the KD/MD mutant VAP sample, and on their MS/MS score (Fig 1D). This strategy led to the identification of 401 proteins, 194 of which were pulled down by both VAP-A and VAP-B. Interestingly, many known partners of VAP-A and VAP-B, such as OSBP, ORP1, ORP2, WDR44, VPS13A, and VPS13C, were identified (Fig 1D). Using a position weight matrix

strategy, we looked for potential FFAT and Phospho-FFAT in the protein sequences; sequences were attributed a score, with 0 corresponding to an ideal FFAT/Phospho-FFAT sequence. Among the 401 proteins identified, 136 had an FFAT or Phospho-FFAT with a significant score (between 0 and 2.5) (Table S1). We used this list of 401 mammalian proteins and identified their *Drosophila* orthologs using DRSC Integrative Ortholog Prediction Tool (Hu et al, 2011), and the fly orthologs with the best score were identified. Using this approach, we were able to identify fly orthologs with more than 90% coverage for 393 of 401 mammalian proteins in the VAP interaction list (Table S3).

## Strategy of a genetic screen

To identify in vivo regulators of RDGB function at ER-PM contact sites, we used a hypomorphic allele rdgB[9] (Vihtelic et al, 1991). rdgB[9] expresses a small amount of residual RDGB protein that provides some function in contrast to the protein null allele rdgB[2]. The FFAT motif of RDGB interacts with the ER-resident membrane protein dVAP-A to provide both localization and function to RDGB (Yadav et al, 2018). FFAT motifs are found in many proteins of varied biological functions and serve to localize them to ER contact sites through a protein–protein interaction with VAP (Murphy & Levine, 2016). We reasoned that if several proteins with an FFAT motif bind to VAP at the ER-PM interface, the lipid transfer function of RDGB could be modulated by their presence at the ER-PM MCS (Fig 2A). Such proteins relevant to RDGB function could be identified by testing their ability to modify the phenotype of the rdgB mutant.

rdgB[9] shows retinal degeneration that is enhanced when flies are grown under illumination (Harris & Stark, 1977; Stark et al, 1983). Under illumination, rdgB[9] flies show severe retinal degeneration by 2 d post-eclosion, making it difficult to score for modulation of this phenotype by other gene products. To overcome this problem, we reared rdgB[9] flies without illumination, a condition under which the retinal degeneration still occurs but at a slower rate; in dark-reared rdgB[9] flies, it takes 2 d for the retinal degeneration to set in, and by day 4, complete retinal degeneration was seen (Fig 2B). Retinal degeneration was scored by visualizing the deep pseudopupil (DPP) under a fluorescence stereomicroscope (Georgiev et al, 2005). To visualize fluorescent pseudopupil, a protein fusion of rhodopsin 1 (Rh1) was tagged with GFP, expressed under its own promoter, and recombined in rdgB[9]. Under these conditions, rdgB[9] shows a clear fluorescent DPP on day 1 that is lost by day 4 with the progression of retinal degeneration (Fig 2Ci).

To identify molecules regulating RDGB function, we depleted their mRNA levels using transgenic RNAi from publicly available collections (Dietzl et al, 2007; Perkins et al, 2015); for 5 of 393 fly genes, there was no RNAi line available from public resources (Table S3). The eye-specific Rh1 promoter was used to restrict GAL4 expression and thus gene depletion, in space to the outer six photoreceptors and in time to post–70-h pupal development (Yadav et al, 2015). To validate the genetic screen, G$_{\alpha q}$ was down-regulated in the rdgB[9] flies and the pseudopupil was scored after days 2 and 4. Knocking down G$_{\alpha q}$ in rdgB[9] flies under Rh1 promoter showed partial suppression of retinal degeneration, and hence, pseudopupil presence after day 4 in dark suggested the efficacy of the screening method (Fig 2Cii).

Using this strategy, we depleted each of the 388 VAP-interacting proteins via RNAi in the rdgB[9]-sensitized background (Fig 2Ciii, Table S3). The screen was performed such that the phenotype arising from off-targets could be minimized. We first used a single RNAi line per gene of interest for the pseudopupil analysis, and once a positive phenotype was scored, the assay was repeated with a second independent RNAi line for the same gene. Only those genes were finally tabulated where two independent lines per gene showed a positive phenotype. To assay the enhancement of retinal degeneration, fly eyes were visualized on day 2, whereas for suppression, fly eyes were checked on day 4. Any suppressor that showed complete recovery of DPP

was scored as a full rescue, whereas others were designated as partial suppressors.

Of 388 genes, knockdown of 52 (two independent RNAi lines per gene) in rdgB[9] showed suppression of retinal degeneration (Fig 2D, Table 1) (Table S4); we designated these as su(rdgB). In this study, we did not identify any candidate that showed enhancement of degeneration when depleted in rdgB[9]. Moreover, 15 genes where only a single RNAi line was available, when tested, did not result in adult progeny (larval death/pupae formed but no fly emerged). Based on their Gene Ontology tags, the 52 su(rdgB) could be classified into several categories (Fig 2E). Of these, the largest number of suppressors was from the class of RNA binding and DNA/chromatin binding proteins. Examples of candidates with strong suppression phenotypes are the pleckstrin homology (PH) domain–containing protein CG9205, phosphorylated adaptor for RNA export (PHAX), ceramide transfer protein (Cert), anaphase-promoting complex 7 protein (APC7), and laminin G domain–containing protein Kon-tiki (Fig 2Ciii). These findings indicate that the mechanisms underlying retinal degeneration in rdgB[9] likely involve diverse sub-cellular processes.

## Identification of suppressors specific to rdgB[9]

In principle, depletion of a gene product can suppress retinal degeneration in rdgB[9] by one of the two mechanisms: (i) by altering the underlying biochemical abnormality resulting from loss of RDGB function, that is, the trigger; and (ii) by down-regulating downstream sub-cellular processes that are part of the degenerative process, that is, the effectors. Genes in the first category, that is, the trigger mechanism, might be expected to suppress only the degeneration of rdgB[9] and no other retinal degeneration, whereas genes that are effectors of retinal degeneration might be expected to suppress multiple retinal degeneration mutants.

To distinguish these two categories of genes, we tested each of the 52 su(rdgB) for their ability to block retinal degeneration in norpA[p24] (Fig 3A and Table S5). norpA encodes for the PLC and catalyses the hydrolysis of PI(4,5)P$_2$ to DAG and IP$_3$. norpA[p24] is a strong hypomorph and shows light-dependent retinal degeneration (Pearn et al, 1996). Of the 52 su(rdgB), 13 genes partially suppressed light-dependent retinal degeneration in norpA[p24], suggesting that they likely participate in the process of retinal degeneration (Fig 3B and C). Most genes in this category belong to the class of RNA binding/processing and DNA/chromatin binding (Fig 3C). The remaining 39 genes therefore likely represent unique suppressors of rdgB[9] and therefore may participate specifically in the trigger mechanism.

## ERG screen to identify su(rdgB) that may regulate phototransduction

A direct test of the role of a candidate in regulating phototransduction will be its ability, when depleted in an otherwise WT fly, to alter the electrical response to light. This can be monitored using ERG that are extracellular recordings that measure the electrical signal from the eye in response to a light stimulus (Vilinsky & Johnson, 2012). Any deviation of the ERG amplitude when compared to that from a WT fly will imply that the interactor likely

**Table 1.** List of *rdgB* interactors.

| Total genetic interactors | Primary accession number (UniProt) | First RNAi line ID | Suppression | Second RNAi line ID | Suppression | Function | Human ortholog | Human primary accession number (UniProt) | Sequence identity with fly homologs (%) | Associated phenotypes | OMIM number |
|---|---|---|---|---|---|---|---|---|---|---|---|
| CG8069 | A1Z7P3 | 100778/KK | ++ | 28189/GD | ++ | Phosphorylated adaptor for RNA export | PHAX | Q9H814 | 29.1 | | 604924 |
| CG4548 | Q9GQN5 | 101568/KK | ++ | 10618/GD | ++ | XNP/adenosinetriphosphatase | ATRX | P46100 | 40.56 | Alpha-thalassaemia/mental retardation syndrome | 300032 |
| CG7961 | Q9W0B8 | 35305/GD | ++ | 35306/GD | + | Coat protein (coatomer) α | COP-A | P53621 | 71.64 | Autoimmune interstitial lung, joint, and kidney disease | 601924 |
| CG7843 | Q9V9K7 | 106344/KK | + | 22574/GD | + | Arsenic resistance protein 2 | SRRT (Isoform 5) | Q9BXP5 | 46.58 | | 614469 |
| CG42665 | Q9VVC6 | 105885/KK | + | 101144/KK | + | Ephexin | ARHGEF5 | Q9BXP5 | 31.12 | Breast cancer | 600888 |
| CG8092 | A0A0B4KER0 | 28196/GD | + | TRiP 25971 | + | Relative of WOC | POGZ (Isoform 5) | Q7Z3K3 | 21.1 | White–Sutton syndrome | 614787 |
| CG42458 | Q7KU81 | 106608/KK | ++ | 108072/KK | ++ | UN, mRNA binding | HNRNPC (Isoform 4) | P07910 | 29.73 | | 164020 |
| CG42522 | Q7KTH8 | TRiP 33370 | ++ | No 2nd RNAi available | | COP9 signalosome subunit 8 | COPS8 (Isoform 2) | Q99627 | 24.73 | | 616011 |
| CG1677 | Q9W3R9 | 109697/KK | ++ | 50195/GD | + | UN, predicted to be involved in mRNA splicing, via spliceosome | ZC3H18 (Isoform 2) | Q9BXP5 | 32.12 | | Not applicable |
| CG14443 | Q9W3Y5 | 105254/KK | ++ | 17618/GD | ++ | UN, RNA helicase | DDX21 | Q9NR30 | 22.99 | | 606357 |
| CG1542 | Q9V9Z9 | 104575/KK | ++ | 39976/GD | ++ | UN, predicted to be involved in rRNA processing and ribosomal large subunit biogenesis | EBNA1BP2 | Q99848 | 42.61 | | 614443 |
| CG9825 | Q9VIZ1 | 105868/KK | ++ | 1712/GD | ++ | UN, solute carrier family 17 (SLC17) member | SLC17A7 | Q13428 | 16.41 | | 605208 |
| CG9205 | Q9W0K9 | 107612/KK | + | 29079/GD | ++ | UN, oxysterol binding protein; PH domain | OSBPL11 | Q9BXB4 | 37.5 | | 606739 |
| CG7971 | A8JNI2 | 101384/KK | ++ | 34262/GD | + | UN, predicted to be involved in RNA splicing | SRRM2 | Q9UQ35 | 29.45 | Intellectual developmental disorder, autosomal dominant 72 | 606032 |
| CG4799 | P52295 | 102627/KK | + | 32466/GD | ++ | Pendulin | KPNA6 | P52292 | 50.58 | | 610563 |
| CG9915 | A8JV07 | 103731/KK | ++ | No 2nd RNAi available | | UN, predicted to be involved in poly(A)+ mRNA export from the nucleus | IWS1 (Isoform 2) | Q96ST2 | 32.63 | | Not applicable |
| CG13849 | Q95WY3 | 103738/KK | + | 51775/GD | + | Nop56 | NOP56 | O00567 | 62.73 | Spinocerebellar ataxia 36 | 614154 |

**Table 1. Continued**

| Total genetic interactors | Primary accession number (UniProt) | First RNAi line ID | Suppression | Second RNAi line ID | Suppression | Function | Human ortholog | Human primary accession number (UniProt) | Sequence identity with fly homologs (%) | Associated phenotypes | OMIM number |
|---|---|---|---|---|---|---|---|---|---|---|---|
| CG9181 | Q9W0G1 | 108888/KK | ++ | 37436/GD | ++ | Protein tyrosine phosphatase 61F | PTPN12 | Q05209 | 26.51 | Colon cancer, somatic | 600079 |
| CG4396 | Q9VYI0 | 101508/KK | + | 48891/GD | + | found in neurons | ELAVL1 | Q15717 | 62.31 | | 603466 |
| CG33967 | Q9VFG8 | 106507/KK | ++ | 100765/KK | ++ | KIBRA | WWC1 | Q8IX03 | 37.1 | Memory, enhanced, QTL | 610533 |
| CG13176 | Q7JIW27 | 39769/GD | ++ | 24642/GD | ++ | Washout | WASH6P | Q9NQA3 | 30.32 | | Not applicable |
| CG3071 | Q9W4Z9 | 107206/KK | ++ | 29589/GD | + | UN, predicted to have snoRNA binding activity | UTP15 | Q8TED0 | 37.35 | | 616194 |
| CG1598 | Q7JWD3 | 110555/KK | + | 32391/GD | ++ | Unnamed/adenosinetriphosphatase | GET3 | O43681 | 68.07 | Cardiomyopathy, dilated, 2H | 601913 |
| CG40218 | Q8SXI2 | 102960/KK | ++ | No 2nd RNAi available | | Yeti | CFDP1 | Q9UEE9 | 31.93 | | 608108 |
| CG4697 | Q9VJR9 | 34308/GD | ++ | 34307/GD | ++ | COP9 signalosome subunit 1a | GPS1 | Q13098 | 37.11 | | 601934 |
| CG14444 | Q9W3Y6 | 110729/KK | ++ | 17622/GD | ++ | Anaphase-promoting complex subunit 7 | ANAPC7 (Isoform 2) | Q9UJX3 | 25.36 | Ferguson–Bonni neurodevelopmental syndrome | 606949 |
| CG2890 | Q9W2U4 | 105399/KK | ++ | 25445/GD | ++ | Protein phosphatase 4 regulatory subunit 2-related protein | PPP4R2 (Isoform 3) | Q9NY27 | 31.66 | | 613822 |
| CG7283 | Q9VTP4 | 109345/KK | ++ | 23459/GD | ++ | Ribosomal protein L10Ab | RPL10A | P62906 | 76.96 | | 615660 |
| CG2807 | Q9VPR5 | 110091/KK | + | 25162/GD | ++ | Splicing factor 3b subunit 1 | SF3B1 | O75533 | 79.95 | Myelodysplastic syndrome, somatic | 605590 |
| CG6538 | P41900 | 110569/KK | + | 12602/GD | + | Transcription factor TFIIFβ | GTF2F2 | P13984 | 50.83 | | 189969 |
| CG18332 | Q8SYG2 | 101516/KK | + | 12821/GD | ++ | COP9 signalosome subunit 3 | COPS3 | Q9UNS2 | 52.84 | | 604665 |
| CG6379 | Q9W4N2 | 103723/KK | + | 29611/GD | ++ | Unnamed/methyltransferase cap1 | CMTR1 | Q8N1G2 | 38.35 | | 616189 |
| CG1427 | Q9VNE3 | 105727/KK | + | 17456/GD | ++ | Sec synthetase | SEPSECS (Isoform 3) | Q9HD40 | 46.43 | Pontocerebellar hypoplasia type 2D | 613009 |
| CG10275 | Q9VJ82 | 106680/KK | ++ | 37283/GD | ++ | Kon-tiki | CSPG4 | Q6UVK1 | 24.52 | | 601172 |
| CG2790 | Q9W0X8 | 101619/KK | + | 20903/GD | + | UN, the heat shock protein 40 (Hsp40) family of co-chaperones | DNAJC21 | Q8N752 | 30.95 | Bone marrow failure syndrome 3 | 617048 |
| CG10415 | O96880 | 100572/KK | + | 12592/GD | + | Transcription factor IIEα | GTF2E1 | P29083 | 46.23 | | 189962 |

| Total genetic interactors | Primary accession number (UniProt) | First RNAi line ID | Suppression | Second RNAi line ID | Suppression | Function | Human ortholog | Human primary accession number (UniProt) | Sequence identity with fly homologs (%) | Associated phenotypes | OMIM number |
|---|---|---|---|---|---|---|---|---|---|---|---|
| CG11228 | Q8T0S6 | 104169/KK | ++ | 7823/GD | + | Hippo | STK3 | Q13188 | 58.45 | | 605030 |
| CG1091 | Q9VI58 | 107175/KK | + | 16088/GD | ++ | Tailor, RNA uridylyltransferase | TUT1 | Q9H6E5 | 23.19 | | 610641 |
| CG33106 | Q9VCA8 | 103411/KK | ++ | 33394/GD | + | mask, multiple ankyrin repeats, single KH domain | ANKRD17 (Isoform 6) | O75179 | 47.77 | Chopra–Amiel–Gordon syndrome | 615929 |
| CG33208 | Q86BA1 | 105837/KK | + | 25371/GD | + | MICAL, molecule interacting with CasL | MICAL3 | Q7RTP6 | 33.76 | | 608882 |
| CG15224 | P08182 | 106845/KK | + | 32377/GD | + | Casein kinase II β subunit | CSNK2B | P67870 | 87.91 | Poirier–Bienvenu neurodevelopmental syndrome | 115441 |
| CG17593 | Q9VQR9 | 106469/KK | ++ | 13029/GD | ++ | UN, orthologous to human CCDC47 (coiled-coil domain containing 47) | CCDC47 | Q96A33 | 43.52 | Trichohepatoneurodevelopmental syndrome | 618260 |
| CG33017 | A1ZAC8 | 103968/KK | + | 40022/GD | + | UN, the MADF-BESS domain transcription regulators | GPATCH8 (Isoform 2) | Q9UKJ3 | 21.17 | | 614396 |
| CG4299 | P53997 | 108987/KK | + | TRiP 77433 | + | Set, encodes a subunit of the inhibitor of the histone acetyltransferase (INHAT) complex | SET | Q01105 | 58.17 | Intellectual developmental disorder, autosomal dominant 58 | 600960 |
| CG7207 | Q9Y128 | 103563/KK | ++ | 27914/GD | + | Ceramide transfer protein | CERT | Q9Y5P4 | 44.31 | Intellectual developmental disorder, autosomal dominant 34 | 604677 |
| CG4202 | Q9I7W5 | 103352/KK | ++ | 49946/GD | + | Something about silencing 10 | UTP3 | Q9NQZ2 | 37.2 | | 611614 |
| CG9977 | Q9VZX9 | 106749/KK | ++ | 36193/GD | ++ | Adenosylhomocysteinase-like 1 | AHCYL1 | O43865 | 72.87 | | 607826 |
| CG32847 | Q8IQM1 | 104294/KK | ++ | 48423/GD | ++ | UN, contains the RING (Really Interesting New Gene) finger domain | TRIM26 | Q12899 | 21.25 | | 600830 |
| CG7839 | Q9VTE6 | 105979/KK | + | 12691/GD | + | UN, orthologous to human CEBPZ (CCAAT/enhancer binding protein zeta). | CEBPZ | Q03701 | 29.87 | | 612828 |
| CG7483 | Q9VHS8 | 108580/KK | ++ | TRiP 32444 | + | eIF4AIII, ATP-dependent RNA helicase | EIF4A3 | P38919 | 86.97 | Robin sequence with cleft mandible and limb anomalies | 608546 |
| CG9556 | Q94899 | 48044/GD | ++ | TRiP 28908 | ++ | Alien | COPS2 | P61201 | 83.97 | | 604508 |
| CG3825 | Q9W1E4 | 107545/KK | + | TRiP 33011 | + | Protein phosphatase 1 regulatory subunit 15 | PPP1R15B | Q5SWA1 | 22.33 | Microcephaly, short stature, and impaired glucose metabolism 2 | 613257 |

Summary of *Drosophila rdgB* genetic interactors identified in the screen. Gene name and/or CG number in FlyBase (www.flybase.org) and UniProt (https://www.uniprot.org/) accession number along with their GO functional annotation. For each gene, the ID of RNAi lines from the VDRC or TRiP library used is shown. Phenotypes scored after depletion of each gene are represented under the "Suppression" column; "++" denotes definite suppression, whereas "+" denotes partial suppression. The human ortholog of each *rdgB* interactor is identified. Known phenotypes associated with each human homolog are denoted along with the Online Mendelian Inheritance in Man (OMIM) identifier number.

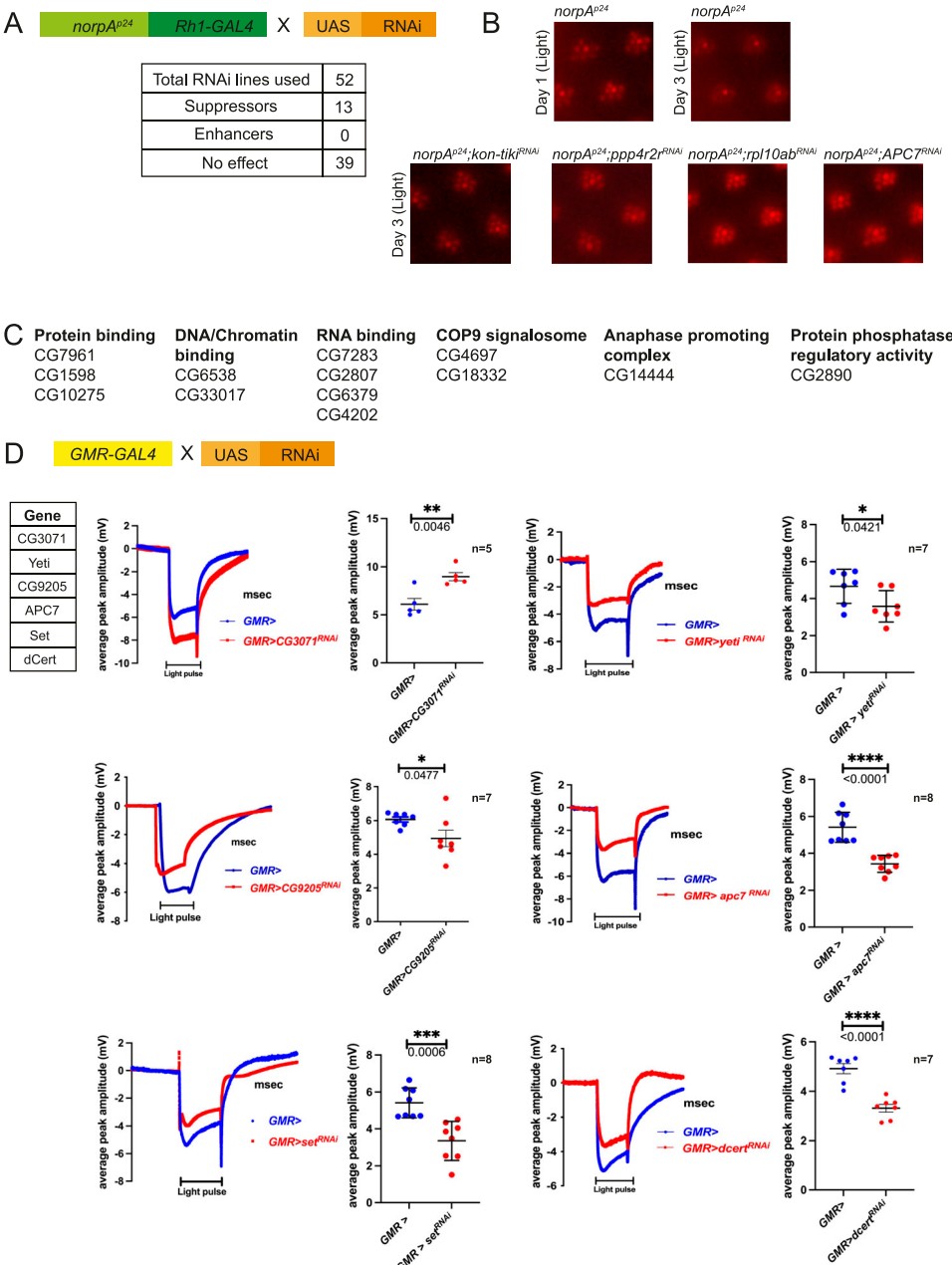

**Figure 3. Genetic screen using *norpA^p24*.**
**(A)** Scheme used to test for genetic interaction of each of the 52 *su(rdgB)* with *norpA^p24* under illumination conditions (constant light 2000 Lux). **(B)** *norpA^p24* flies degenerate by day 3 under light conditions, and examples of *su(RDGB)* candidates that suppressed the *norpA^p24* retinal degeneration phenotype. n = 5 flies/RNAi line. **(C)** Complete list of 13 genes with their cellular functions that suppressed the *norpA^p24* phenotype. ERG screen. **(D)** Of 52 candidates, five *su(RDGB)* showed reduced (CG9205, Yeti, Apc7, Set, and dCert) and one (CG3071) showed higher ERG phenotype (traces and quantification shown) when down-regulated in an otherwise WT background. The number of flies used for the experimental set is mentioned along with the quantification. Scatter plots with the mean ± SEM are shown. Statistical tests: unpaired *t* test.

functions in the process of phototransduction. We down-regulated each *su(rdgB)* using the eye-specific promoter, GMR-GAL4, in an otherwise WT background and measured ERG amplitudes. Of 52 *su(rdgB)*, GMR-driven knockdown (in both of two independent RNAi lines) of five candidates (CG9205, Yeti, APC7, Set, and Cert) showed a lower ERG amplitude and of one candidate (CG3071) a higher ERG amplitude compared with control flies (Fig 3D and Table S6). In the case of six additional *su(rdgB)*, depletion with GMR-GAL4 resulted in a rough eye phenotype with the first RNAi line (Fig S1Ai). When a second independent RNAi line was used, four (Ars2, CG7483, Cmtr1, and Secs) of six candidates showed lower ERG amplitude (Fig S1Aiii). A rough eye phenotype after knocking down Rpl10ab and Sf3b1 with

multiple RNAi lines points towards involvement of these genes in the eye development (Fig S1Aii).

## The spatial and temporal profile of dCert and CG9205 down-regulation results in contrasting impact on *rdgB^9* phenotypes

To confirm the findings of our RNAi depletion studies, we sought to study the impact of germline mutations in candidates identified in the RNAi screen on *rdgB^9*. Of the six genes identified as specific *rdgB* interactors, *CG9205, yeti, APC7, set, dcert,* and *CG3071*, there were no mutants available in two of them (*set* and *CG3071*). Mutants in *yeti* are homozygous lethal, making it difficult to work on it in this

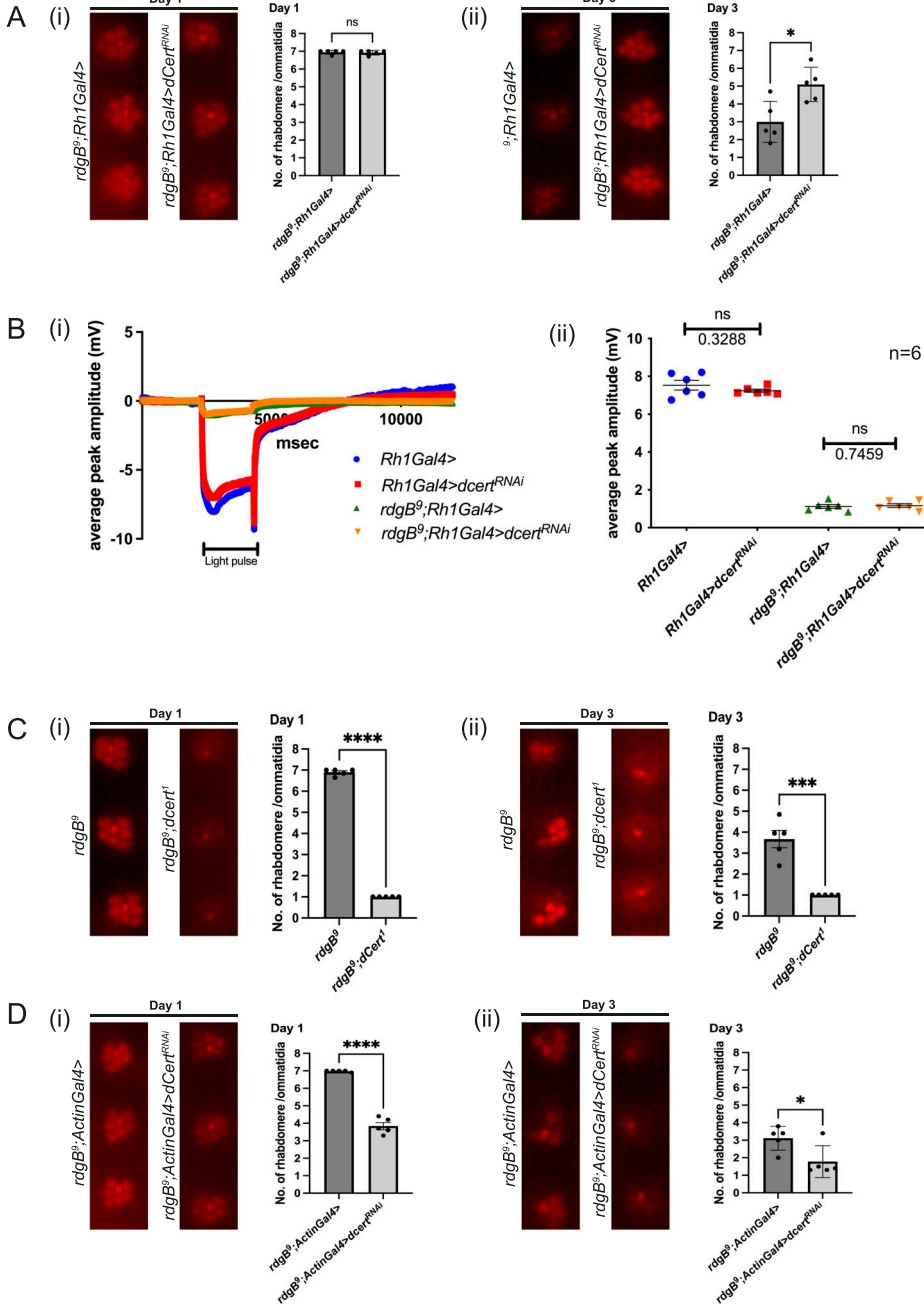

**Figure 4. Spatial and temporal down-regulation of dCert in *rdgB⁹*.**

**(A)** Suppression of retinal degeneration when dCert RNAi line (35579/TRiP, BDRC) was expressed using Rh1 promoter. After eclosion, flies were kept in the dark and assayed on either day 1 or 3: (i) on day 1, there was no appreciable difference in two genotypes and rhabdomeres were intact; and (ii) on day 3, down-regulation of dCert in *rdgB⁹* suppressed the retinal degeneration observed in *rdgB⁹* control. **(B)** When subjected to ERG analysis, down-regulation of dCert using Rh1-GAL4 in the background of *rdgB⁹* did not suppress the ERG phenotype: (i) ERG trace and (ii) quantification. n = 6 flies. Scatter plots with the mean ± SEM are shown. Statistical tests: unpaired *t* test. **(C)** Double mutant of *rdgB⁹;dcert¹* showed enhancement of retinal degeneration: (i, ii) by day 1 alone, double mutant has severely enhanced retinal degeneration phenotype when compared to *rdgB⁹*. **(D)** Enhancement of retinal degeneration when dCert (35579/TRiP, BDRC) was down-regulated with a whole-body Actin-Gal4 promoter in the *rdgB⁹* background: (i) on day 1, rhabdomere loss is significant in the experimental flies compared with control that worsens by day 3 and phenocopies the retinal degeneration present in the double mutant. For optical neutralization experiments, scoring was done by quantifying 10 ommatidia/fly head, n = 5 fly heads.

setting. A viable mutant in *APC7* is available, but the encoded protein has no obvious membrane interaction domains. However, as dCert and CG9205 have membrane-interacting domains, we chose to focus on these two candidates for the proof-of-principle analysis. As previously noted, down-regulation of dCert in *rdgB⁹* caused suppression of retinal degeneration when the RNAi construct was expressed using the Rh1 promoter (Fig 4Ai and ii), although the suppression of retinal degeneration was not sufficient to rescue the ERG phenotype of *rdgB⁹* (Fig 4Bi and ii). We retested this genetic interaction using a germline mutant allele of *dcert* (*dcert¹*) (Rao et al, 2007). Surprisingly, the double mutant *rdgB⁹;*

*dcert¹* showed enhancement of retinal degeneration compared with *rdgB⁹* (Fig 4Ci and ii). We confirmed these findings using the same dCert RNAi line used in the screen (expressed using Rh1 Gal4) but this time with the whole-body expression of the RNAi using Actin-GAL4 that expresses throughout development beginning with embryogenesis. In *rdgB⁹;actin>dcert^RNAi*, we found enhancement of retinal degeneration such that by day 3, all photoreceptors except R7 were completely degenerated (Fig 4Di and ii), thus recapitulating the observations seen with *rdgB⁹; dcert¹*. In a similar fashion, down-regulating CG9205, which encodes a PH domain–containing protein, under the Rh1 promoter

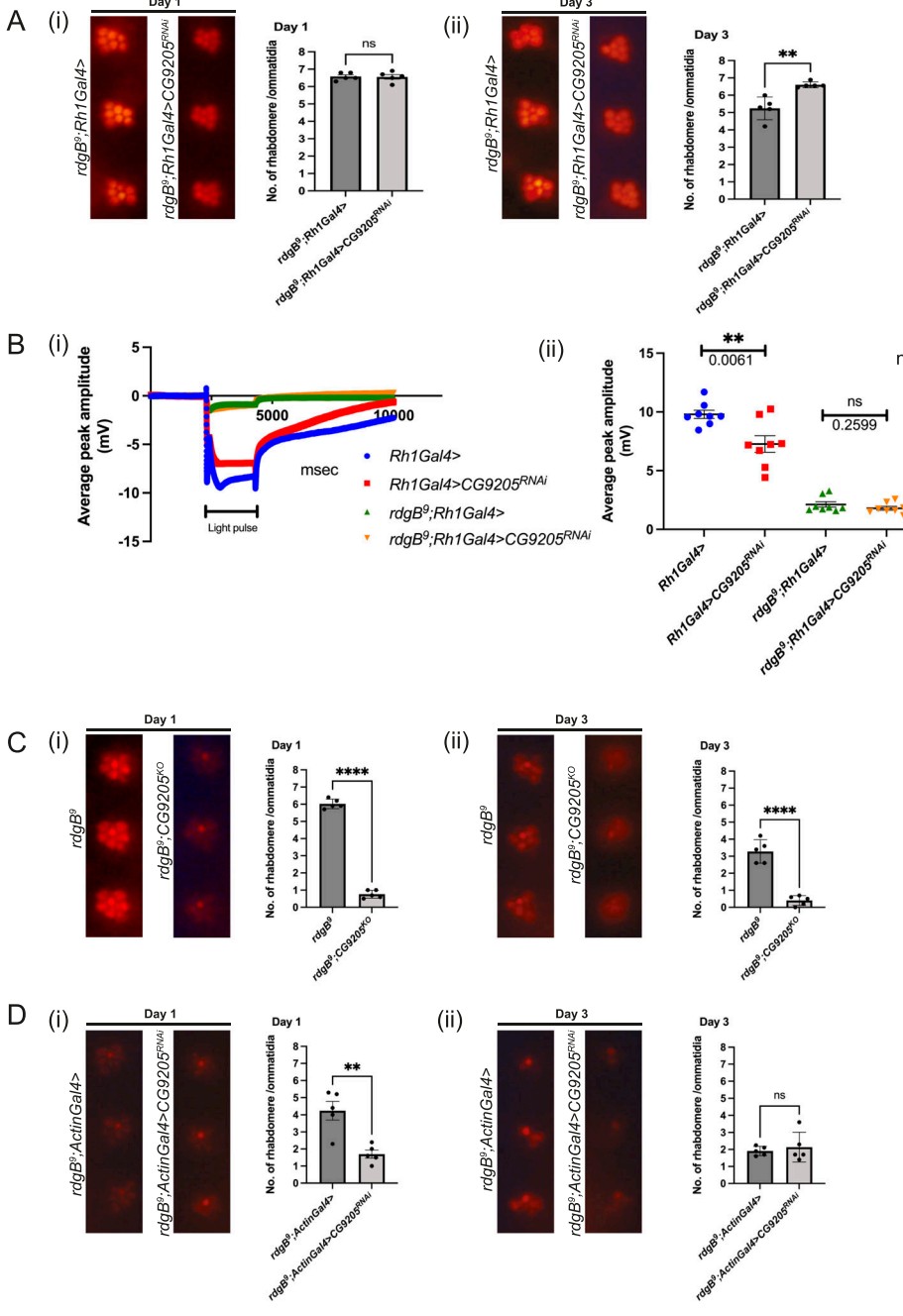

**Figure 5. Spatial and temporal down-regulation of CG9205 in *rdgB⁹*.**
**(A)** Suppression of retinal degeneration when CG9205 RNAi line (29079/GD, VDRC) was expressed using Rh1 promoter. After eclosion, flies were kept in the dark and assayed on either day 1 or 3: (i) on day 1, there was no appreciable difference in two genotypes and rhabdomeres were intact; and (ii) on day 3, down-regulation of CG9205 in *rdgB⁹* suppressed the retinal degeneration observed in *rdgB⁹* control. **(B)** When subjected to ERG analysis, down-regulation of CG9205 using Rh1-GAL4 in the background of *rdgB⁹* did not suppress the ERG phenotype, whereas down-regulation of CG9205 using Rh1-GAL4 in an otherwise WT background shows reduced ERG amplitude: (i) ERG trace and (ii) quantification. n = 8 flies. scatter plots with the mean ± SEM are shown. Statistical tests: unpaired *t* test. **(C)** Double mutant of *rdgB⁹;CG9205^{KO}* showed enhancement of retinal degeneration: (i, ii) by day 1 alone, double mutant has severely enhanced retinal degeneration phenotype when compared to *rdgB⁹*. **(D)** Enhancement of retinal degeneration when CG9205 (29079/GD, VDRC) was down-regulated with a whole-body Actin-Gal4 promoter in the *rdgB⁹* background: (i) on day 1, rhabdomere loss is significant in the experimental flies compared with control that remains the same on day 3 and phenocopies the retinal degeneration present in the double mutant. For optical neutralization experiments, scoring was done by quantifying 10 ommatidia/fly head, n = 5 fly heads.

resulted in the suppression of retinal degeneration in *rdgB⁹* (Fig 5Ai and ii). When subjected to ERG, the suppression of retinal degeneration did not result in ERG rescue, whereas in an otherwise WT background, reduction in CG9205 levels under Rh1 showed a reduced ERG amplitude (Fig 5Bi and ii). In contrast, when a germline CG9205 CRISPR knockout (*CG9205^{KO}*) was combined with *rdgB⁹*, there was enhanced photoreceptor degeneration (Fig 5Ci and ii). This result was corroborated using the whole-body promoter, Actin-Gal4, to reduce CG9205 levels in the *rdgB⁹* background where retinal degeneration was enhanced (Fig 5Di and ii). These findings suggest that dCert and CG9205 depletion more broadly in the fly across both space and time

domains may have distinctive effects compared with a more restricted expression in post-mitotic adult photoreceptors using Rh1 GAL4. This finding also suggests multiple modes of action for dCert and CG9205 in *Drosophila* photoreceptors during distinct phases of photoreceptor development.

# Discussion

Neurodegeneration is a complex process involving multiple layers of cellular and molecular events leading to the phenotype

observed in vivo. Regardless of the part of the nervous system that is affected, be it the central or peripheral, conceptually, the processes leading to any neurodegeneration can be classified into two groups: (i) trigger steps—that is, those initial molecular or biochemical changes that initiate the process of degeneration; and (ii) effector steps—that is, those steps that are subsequently part of the process that leads to loss of neuronal structure and consequently function. Identifying the molecular processes involved in each of these steps is critical for developing strategies to manage neurodegenerative disorders. The *Drosophila* eye has been used in several settings for modelling neurodegeneration (Bonini & Fortini, 2003) such as those caused by repeat disorders such as Huntington's disease and various ataxias, Alzheimer's disease, and primary degenerative disorders of the human retina (Xiong & Bellen, 2013). In the present study, we performed a genetic analysis to uncover the mechanisms of retinal degeneration underlying mutants in *rdgB*, which encodes a Class II PITP. Mutations in Class I PITP (PITPα) in mice result in a neurodegeneration phenotype (Hamilton et al, 1997), and recently, human patients carrying mutations in VPS13 have been reported with neurodegenerative disorders (Ugur et al, 2020). Thus, the findings of our screen will inform on mechanisms of neurodegeneration.

The ER-resident protein VAP that has been linked to neurodegenerative diseases (Nishimura et al, 2004) is known to interact with multiple cellular proteins via the binding of its MSP domain with the FFAT motif of other proteins (Murphy & Levine, 2016). In this study, we performed an immunoprecipitation analysis that exploits this FFAT-VAP interaction and were able to identify 401 proteins enriched in an IP-MS experiment. Surprisingly, only 136 of these proteins had an identifiable FFAT or Phospho-FFAT motif. This is despite the list of 401 proteins being curated as those enriched in an IP-MS with WT VAP but not a non-FFAT binding version of this protein. This implies that 265 proteins (66%) of the total proteins identified in the IP-MS experiment do not interact directly with VAP but do so indirectly presumably via other proteins (Fig 2A). This suggests that VAP-interacting proteins in cells include both categories (a smaller direct FFAT-VAP–meditated set and a larger indirect interactor set) and both groups may influence molecular processes in the vicinity of VAP. How significant are the interactions of VAP with these non–FFAT-containing proteins identified in the IP-MS proteomics experiment? When tested experimentally, we found that one such protein CG9205 could interact with VAP in an immunoprecipitation experiment from *Drosophila* head extracts (Fig S2). In contrast, another such non–FFAT-containing interactor casein kinase II *β* did not interact with VAP under similar conditions (data not shown). Interestingly, a previous large-scale experimental study of cellular protein–protein interactions that described a VAP interactome through mass spectrometry–based methods also noted non–FFAT-containing proteins such as LSG1 identified as interactors of VAP in mammalian cells (Huttlin et al, 2015). Of 38 VAP interactors identified in the BioPlex study, only 12 have identifiable FFAT motifs. These findings highlight the possibility that VAP-interacting proteins identified using proteomics screens may also include both direct and indirect interacting groups. It also suggests that although bioinformatics-based identification of FFAT motifs in sequenced genomes from other systems will continue to be a useful means of identifying VAP-interacting proteins,

experimental methods such as IP-MS will add to the identification of the larger group of VAP-interacting proteins, many of whom may not have FFAT motifs.

To understand the cellular and molecular processes underlying retinal degeneration in *rdgB⁹*, we depleted selected molecules using RNAi and scored for suppression of retinal degeneration. The candidates selected for screening were initially identified using an IP-MS proteomics screen for interactors of VAP-A and VAP-B in cultured mammalian cells; however, the functional significance of their interaction with VAP was not known. Although previous studies have identified many VAP-interacting proteins in mammalian cell culture models by protein interaction studies, the functional relevance of these for in vivo function and neurodegeneration remains unknown. Using our in vivo analysis, we were able to identify a subset (52 of 388) of these interactors in our proteomics screen that, when depleted, suppressed the retinal degeneration in *rdgB⁹*. This finding underscores the value of an in vivo genetic screen in evaluating the functional effect of candidates identified in vitro to understanding the mechanisms of neurodegeneration. The human homologs in 13 of the *su(rdgB)* genes have previously been linked to human neurodevelopmental or neurodegenerative disorders (Table 1), and a large proportion of the 52 *su(rdgB)* have human homologs that show high expression in the human brain (Table S7). Thus, the findings of this study could provide important insights into the mechanisms of human brain disorders.

Because our primary screen for suppressors of *rdgB* would identify molecules involved in both the trigger and effector steps of the degeneration process, it is essential to classify the identified suppressers into these two categories. Because *rdgB* mutants are known to affect photoreceptor physiology before the onset of retinal degeneration (Yadav et al, 2015), we reasoned that suppressors that work at the level of the trigger might also affect the electrical response to light, the physiological output of the photoreceptor. By this rationale, we found that 6 of 52 suppressors when depleted in an otherwise WT background led to an altered electrical response to light; these suppressors are therefore likely to impact the processes by which RDGB functions in phototransduction. Examples of these include CG9205, Yeti, APC7, Set, Cert, and CG3071. Two of these genes *CG9205* (PH domain–containing) and *cert* (ceramide transfer protein) encode proteins with either ion binding or lipid transfer function, and their ability to act as *su(rdgB)* may indicate a role of previously unidentified lipids and lipid transfer at MCS in phototransduction. In contrast, Set (subunit of the INHT complex that regulates histone acetylation), Yeti (a chromatin-associated protein that interacts with the Tip60 chromatin remodelling complex), and CG3071 (snoRNA that positively regulates transcription by RNA polymerase 1) all likely exert their effect as *su(rdgB)* by modulating gene expression; some of the genes so regulated may impact phototransduction. A transcriptome analysis of *rdgB⁹* photoreceptors may help identify the relevant genes and the manner in which they regulate phototransduction.

To identify molecular mechanisms that regulate the effector steps of the degeneration process, we determined which of the *su(rdgB)* could also suppress another retinal degeneration mutant, *norpA^{p24}*. Such *su(rdgB)* will likely represent molecules that participate in common effector steps of retinal degeneration shared by

these two mutants. The 13 genes so identified represent several different functional classes. Prominent among these classes are RNA binding and DNA/chromatin binding proteins. Overall, a large percentage of *su(rdgB)* identified in our screen were of the class of RNA processing (CG1677, CG1542, CG7971, Cmtr1, Srrm234, Nop56, CG3071, Rpl10, Ars2, CG42458, SecS, CG9915, Sf3b1), RNA editing (Tailor, Sas10), RNA export (Phax), and RNA helicases (CG14443, CG7483). Interestingly, a role of RNA binding proteins such as Ataxin-1 has been proposed in neuronal homeostasis and neurodegenerative processes and our finding may reflect a more general role of RNA binding/homeostasis in neurodegenerative processes (Prashad & Gopal, 2021). A further large group of *su(rdgB)* belong to those regulating transcription (XNP, Fne, Yeti, TFIIFβ, TFIIEα, CG33017, Set, CG7839), and Sf3b1, Cmtr1, Rpl10Ab, TFIIFβ, and Sas10 were among those candidates that in addition suppressed retinal degeneration in *norpA^{p24}*. This finding suggests that regulated transcription may be important for maintaining neuronal homeostasis; this may be particularly significant because neurons are post-mitotic and transcriptional process and RNA turnover may collectively be key mechanisms for maintaining cellular homeostasis.

A third class of *su(rdgB)* were subunits of the COP9 signalosome (CSN1a, CSN2, CSN3, and CSN8 were identified in our screen). The COP9 signalosome acts as a signalling platform regulating cellular ubiquitylation status. The COP9 signalosome has been shown to play a key role in regulating *Drosophila* development through E3 ubiquitin ligases by deNEDDylation (Freilich et al, 1999). In addition, two E3 ubiquitin ligase family members were also identified in the genetic screen: (i) APC7, which is a subunit of anaphase-promoting complex/cyclosome that comprises seven other subunits and is required to modulate cyclin levels during cell cycle; and (ii) *CG32847*, an uncharacterized gene belonging to the "Other RING domain ubiquitin ligases" family of proteins. Ubiquitination could regulate the structure and function of proteins required for phototransduction; depletion of APC7 resulted in a reduction in the ERG amplitude supporting this mechanism. Alternatively, it is possible that ubiquitination-regulated protein turnover may be part of the process of retinal degeneration. Interestingly, a key role of ubiquitination has been described in the context of neurodegeneration (Schmidt et al, 2021).

Overall, our screen uncovers a role of multiple molecular processes regulated by VAP-interacting proteins that are required for maintaining lipid turnover and neuronal homeostasis in photoreceptors. It is important to note that our screen focused on VAP-interacting proteins, but there will also be non–VAP-dependent processes that also contribute to lipid and neuronal homeostasis in photoreceptors. Alternative genetic screens will be required to map their role in photoreceptor maintenance. Collectively, such studies will help advance our understanding of neurodegeneration in the context of LTP function.

# Materials and Methods

### Protein pull-down and mass spectrometry analysis

Recombinant protein expression in *E. coli* and purification using plasmids encoding the MSP domain of VAP-A (8–212; WT and KD/MD mutant) and VAP-B (1–210; WT and KD/MD mutant) were previously described (Di Mattia et al, 2020). For protein pull-down, the affinity resin was prepared by incubating 100 µg of recombinant protein with 20 µl of nickel beads (PureProteome Nickel Magnetic Beads; Merck) in pull-down buffer PDB (50 mM Tris–HCl, pH 7.4, 50 mM NaCl, 1 mM EDTA, 1% Triton X-100, 5 mM imidazole, cOmplete protease inhibitor cocktail [Roche], and PhosSTOP [Roche]). The beads were then washed three times with the same buffer. $8 \times 10^8$ HeLa cells were washed with 5 ml of TBS and lysed with 1 ml of PDB. After a 10-min incubation on ice, the protein extract was separated from cell debris by centrifugation (10 min; 9,500*g*; 4°C). The protein extract was mixed with VAP-coupled nickel beads and incubated for 2 h at 4°C under constant agitation. The beads were then washed three times with PDB, and proteins were eluted with Laemmli buffer. Proteins were precipitated with trichloroacetic acid and digested with Lys-C (Wako) and trypsin (Promega). Peptides were then analysed using Ultimate 3000 nano-RSLC (Thermo Fisher Scientific) coupled in-line with Orbitrap ELITE (Thermo Fisher Scientific).

### SDS–PAGE, Western blot, and Coomassie blue staining

SDS–PAGE and Western blot analysis were performed as previously described (Alpy et al, 2005) using the following antibodies: rabbit anti-STARD3NL (1:1,000; pAbMENTHO-Ct-1545 [Alpy et al, 2001]), rabbit anti-ORP1 (1:1,000; ab131165; Abcam), and mouse anti-actin (1:5,000; A1978; Merck). Coomassie blue staining was performed with PageBlue Protein Staining Solution (Thermo Fisher Scientific).

### In silico identification of potential conventional and Phospho-FFAT motifs

The FFAT scoring algorithm used for Phospho-FFAT identification is based on the position weight matrix (Di Mattia et al, 2020). For conventional FFAT sequences, the Phospho-FFAT matrix described in Di Mattia et al was modified in position 4 to assign a score of 4 to S and T, and a score of 0 to D and E. These algorithms assign conventional and Phospho-FFAT scores to protein sequences. They are based on 19 continuous residues: six residues upstream, seven residues forming the core, and six residues downstream. An ideal sequence scores zero.

### Fly culture and stocks

Flies (*Drosophila melanogaster*) were reared on standard cornmeal, dextrose, yeast medium at 25°C and 50% relative humidity in a constant-temperature laboratory incubator. There was no internal illumination within the incubator, and flies were subject to brief pulses of light only when the incubator doors were opened. To study light-dependent degeneration, flies were exposed to light in an illuminated incubator at an intensity of 2000 lux. *rdgB^9*, P[w+,Rh1::GFP]; Rh1-Gal4, UAS-Dicer2 and *norpA^{p24}*; Rh1-Gal4, UAS-Dicer2 were the strains used for the genetic screens. GMR-Gal4 (second chromosome) was used for ERG experiments. UAS-RDGB, UAS-dCert::TurboID, 3HA-CG9205 and CKIIβ-3HA were used in co-IP experiments and generated in our laboratory.

### Fluorescent DPP analysis

Pseudopupil analysis was carried out on flies after days 2 and 4 post-eclosion. Flies were immobilized using a stream of carbon dioxide, and fluorescent pseudopupil analysis was carried out using an Olympus SZX12 stereomicroscope equipped with a fluorescent light source and GFP optics. Images were recorded using an Olympus digital camera.

### Optical neutralization

Flies were immobilized by cooling on ice. They were decapitated using a sharp razor blade and fixed on a glass slide using a drop of colourless nail varnish. The refractive index of the cornea was neutralized using a drop of immersion oil ($n$ = 1.516 at 23°C); images were observed using a 40× oil-immersion objective (UPlanApo, 1.00 Iris; Olympus) with antidromic illumination (Franceschini & Kirschfeld, 1971). Images were collected on an Olympus BX-41 upright microscope and recorded using an Olympus digital camera.

### Electroretinogram recordings

Flies were anaesthetized and immobilized at the end of a disposable pipette tip using a drop of low melt wax. Recordings were done using glass microelectrodes filled with 0.8% wt/vol NaCl solution. Voltage changes were recorded between the surface of the eye and an electrode placed on the thorax. After fixing and positioning, flies were dark-adapted for 6 min. ERG was recorded with 1-s flashes of green light stimulus. Stimulating light was delivered from an LED light source within 5 mm of the fly's eye through a fibre-optic guide. Voltage changes were amplified using a DAM50 amplifier (WPI) and recorded using pCLAMP 10.2. Analysis of traces was performed using Clampfit (Axon Laboratories).

### Co-immunoprecipitation (fly heads)

Fly heads with respective genotypes were lysed in ice-cold protein lysis buffer (50 mM Tris–HCl, 1 mM EGTA, 1 mM EDTA, 1% Triton X-100, 50 mM NaF, 0.27 M sucrose, and 0.1% $\beta$-mercaptoethanol). 10% of the lysate was aliquoted to be used as input. The remaining lysate was split into two equal parts. To one part, dVAP-A antibody (a kind gift from Girish Ratnaparkhi, IISER Pune) was added, and to the other part, a corresponding amount of control IgG (2729S; CST) was added, and incubated overnight at 4°C. On the next day, protein G Sepharose beads (GE Healthcare) were spun at 130,00$g$ for 1 min and then washed with TBS twice. The beads were then incubated with 5% BSA (HiMedia) in TBS with 0.1% Tween-20 (TBST) for 2 h at 4°C. Equal amounts of blocked beads were then added to each sample and incubated at 4°C for another 2 h. The immunoprecipitates were then washed twice with TBST containing $\beta$-mercaptoethanol and 0.1 M EGTA for 5 min. The supernatant was then removed, and the beads were boiled in 2X Laemmli sample buffer for Western blotting. Primary antibodies used were as follows: rat anti-RDGB (laboratory-generated, 1:4,000), mouse anti-V5 (R960-25, 1:10,000; Invitrogen), and mouse anti-HA (1:1,000; CST). Respective secondary antibodies were used at the dilution of 1:10,000.

## Supplementary Information

## Acknowledgements

This work was supported by the Department of Atomic Energy, Government of India, under Project Identification No. RTI 4006, DBT-Wellcome India Alliance Senior Fellowship to P Raghu (IA/S/14/2/501540), and DBT-Wellcome India Alliance Early Career Fellowship to S Mishra (IA/E/17/1/503653). We thank the NCBS Imaging and Drosophila facilities for their support. We thank Catherine Tomasetto and the other members of the Molecular and Cellular Biology of Breast Cancer team for their helpful advice and discussions. We thank the IGBMC cell culture facility and proteomics platform (Luc Negroni, Frank Ruffenach, and Bastien Morlet) for their excellent technical assistance. This work was supported by grants from the Agence Nationale de la Recherche (ANR) (grant ANR-19-CE44-0003; https://anr.fr/). This work of the Interdisciplinary Thematic Institute IMCBio, as part of the ITI 2021-2028 Program of the University of Strasbourg, CNRS, and Inserm, was supported by IdEx Unistra (ANR-10-IDEX-0002) and by SFRI-STRAT'US Project (ANR 20-SFRI-0012) and EUR IMCBio (ANR-17-EURE-0023) under the framework of the French Investments for the Future Program.

### Author Contributions

S Mishra: data curation, formal analysis, investigation, methodology, and writing—original draft, review, and editing.
V Manohar: investigation.
S Chandel: investigation.
T Manoj: investigation.
S Bhattacharya: investigation.
N Hegde: investigation.
VR Nath: investigation and review.
H Krishnan: formal analysis, investigation, methodology, and writing—original draft, review, and editing.
C Wendling: investigation.
T Di Mattia: investigation.
A Martinet: investigation.
P Chimata: investigation.
F Alpy: conceptualization, data curation, investigation, and writing—original draft, review, and editing.
P Raghu: conceptualization, supervision, funding acquisition, project administration, and writing—original draft, review, and editing.

### Conflict of Interest Statement

The authors declare that they have no conflict of interest.

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
