## [Reviewer comments · Life Science Alliance]

Life Science Alliance

A genetic screen to uncover mechanisms of lipid transfer protein function at membrane contact sites

Shirish Mishra, Vaishnavi Manohar, Shabnam Chandel, Tejaswini Manoj, Subhodeep Bhattacharya, Nidhi Hegde, Vaisaly Nath, Harini Krishnan, Corinne Wendling, Thomas Di Mattia, Arthur Martinet, Prasanth Chimata, Fabien Alpy, and Padinjat Raghu
DOI: <https://doi.org/10.26508/lsa.202302525>

Corresponding author(s): Padinjat Raghu, National Centre for Biological Sciences

Review Timeline:	Submission Date:	2023-12-13
	Editorial Decision:	2023-12-15
	Revision Received:	2024-02-06
	Editorial Decision:	2024-02-22
	Revision Received:	2024-03-04
	Accepted:	2024-03-05

Transaction Report:

Please note that the manuscript was reviewed at *Review Commons* and these reports were taken into account in the decision-making process at *Life Science Alliance*.

Review
COMMONS

Reviews

Review #1

Mishra and colleagues have conducted a large genetic screen to identify modulators of a *Drosophila* model for retinal degeneration. Using biomolecular techniques, they selected a few hundred proteins that interact with an ER bound protein VAP, to further test them in the retinal degeneration model. This was done by downregulating their expression using interference RNA (RNAi). Degeneration was first measured through a pseudopupil analysis, then suppressors of degeneration were further tested in a different retinal degeneration model, and finally through an ERG experiment. Finally, they focused on a strong suppressor of degeneration, dCert, by using a mutant allele to confirm the findings from the RNAi.

The results suggest that a handful of these candidates are suppressors in this model of retinal degeneration, and identify at which stage of retinal degeneration these proteins may be involved in. These proteins may have a significance in forms of neurodegeneration.

****Major comment:****

- Perhaps a larger number of replicates could be done in the optical neutralization experiment as well as in the ERG. Figure 4.A(i) and (ii), please clearly state n values. I would suggest this as optional, but perhaps it could help to increase n?
- For human orthologs (Table 1), it could be worthwhile to add alignment scores between fly and human?

****Minor comment:****

- Clarify the purpose in focusing on dCert specifically in the last results section and discussion
- Several typos
- Affect vs effect

General assessment:

The main significance of this study comes from the focus on proteins that are known to interact with VAP. This implies that the suppressors of degeneration that they have identified in the RdgB9 model may have an effect in other neurodegenerative models, namely in ALS models. This could have a very high significant potential in therapeutic avenues for neurodegenerative diseases.

Among six candidates that had an effect with knocked down through RNAi, they pursued a single one (dCert) as proof of principle. It would help to add a justification for this choice in the main text and whether the authors have performed or intend to perform experiments using mutant forms of the other candidate proteins.

The work from Raghu and his team have been leading the research surrounding this model of degeneration in *Drosophila*. This study naturally further extends their field of research, identifying more candidates that modulate this form of degeneration, and helping elucidate the pathways leading to cellular degeneration.

These results will be of high interest for specialized researchers studying the molecular pathways that lead to cellular degeneration, both in the context of retinal degeneration as well as neurodegeneration. Specifically, researchers that may be interested in these candidate proteins and how they may play a role in the pathogenesis of various degenerative diseases.

Review #2

****Summary:**** In this study, Mishra and colleagues combine proteomic techniques with *Drosophila* genetics to identify interactors of the endoplasmic reticulum (ER) protein VAP and analyse their indirect implication in the

light-dependent retinal degeneration phenotype caused by misfunction of the lipid transfer protein RdgB, another VAP interactor. This is an ideal model system to study neurodegeneration in vivo via which the authors aim to further understand the molecular mechanisms involved in VAP-RdgB's function in maintenance of membrane lipid homeostasis. Of an initial list of 403 VAP mammalian interactors found via Immunoprecipitation - Mass Spectrometry performed in a human cell line, 52 homologous Drosophila genes are found to suppress the RdgB degeneration phenotype upon knockdown. The authors then test a series of genetic interactions to dissect the potential role of these genes in the degeneration phenotype and identify six genes that likely suppress the phenotype by directly acting on the same molecular processes as RdgB, two of which (Cert and CG9205) could have a direct mechanistic role in the lipid homeostasis maintained by the VAP-RdgB complex.

****Major comments:****

- The model suggested by the authors in Fig 2A is one by which the tested genes influence the VAP-RdgB function via direct binding to VAP. The direct interaction with VAP is a core aspect of the study as the IP-MS experiment biases the list of genes tested in Drosophila, and this list is often referred to as "VAP interacting proteins". However, the interactions between the proteins coded by the most relevant genes in the genetic screen and VAP are not tested. In fact, of the six genes that are found to likely modulate the same processes as RdgB, four probably do so via affecting gene expression, as discussed by the authors, therefore making it unlikely that they are true VAP interactors (unless they shuttle from the ER to the nucleus). Additionally, it would seem that most of the 52 genes found to suppress the degeneration phenotype are not necessarily VAP interactors either as only a handful of genes in this list have predicted strong FFAT motifs. In general, the authors should provide additional comments/evidence on the interaction, or likelihood of the interaction, of these proteins with VAP, that should include:

- The IP-MS data should be made available (at least on my end, in the current submission the supplementary lists available cover only the 52 genes found to suppress the phenotype, I found the list of 403 genes in the bioRxiv submission but this also does not include data on the enrichment of each hit in the IP-MS). It should be made clear how enriched were each of the proteins in the analysis, and how high the most relevant genes in the genetic screen rank within this interactor list.

- The authors should provide further detail on the rationale behind defining the list of 403 genes to be tested (for example, what was the threshold enrichment considered as interaction). Also in relation to this, the authors should at least provide speculation as to why more than half of the 403 genes defined do not have an FFAT motif, despite the fact that the proteomic data was normalized to a non-FFAT motif binding mutant of VAP which should be capable of maintaining non-FFAT mediated interactions of the protein.

- The authors should acknowledge the limitations of their experimental design in regards to identifying real interactors of VAP in Drosophila and avoid referring to this set of genes as "VAP interacting proteins" and rather use a more accurate description such as "proteins enriched in the IP-MS" or at least "potential VAP interacting proteins".

- Testing interaction between VAP and all the 52 genes found to suppress the phenotype would be a huge amount of work. But the finding most relevant to the initial premise of the study (i.e. "molecular mechanisms underlying lipid transfer protein function at membrane contact sites") is that Cert (a lipid transfer protein) and CG9205 (fly homolog of a mammalian lipid transfer protein) influence RdgB function. Demonstrating an interaction between these proteins and VAP would argue for the experimental design and support the hypothesis and model of the study. Cert is already a well established interactor of VAP, hence the authors would not need to add anything regarding this protein. Is CG9205 expected to be also a true interactor of VAP? Biochemical experiments could be used to test this idea, or even recently developed in silico modelling of interactions (i.e. AlphaFold Multimer) could be of help. If no interaction is observed/expected this should also be pointed out in the manuscript. Optionally, showing localization of Cert or CG9205 to the ER-PM interface would also greatly support the model of VAP-RdgB regulation suggested by the authors.

****Minor comments:****

- In the model shown in Fig2A, it would seem that many proteins can bind VAP in addition to RdgB, however, VAP proteins have only one FFAT binding pocket. This model would only be possible if oligomerization of VAP is considered (oligomerization of VAP has been reported to occur, see for example PMID:20207736). The model should be redrawn considering this fact.

- In line 161 of the text VPS13D is mentioned, however VPS13C is the gene indicated in Fig 1D.

Previous studies by some of the authors and others have shown that RdgB can transfer lipids between the ER, to which it binds via VAP, and the plasma membrane (PM), and is required for proper replenishment of PI(4,5)P₂ in the PM which is in turn necessary for sustained PLC signaling in *Drosophila* photoreceptors. Lack of RdgB leads to light-dependent degeneration of the retina, and hence it is utilized by the authors as a model for neurodegeneration. Given the clear phenotype of RdgB loss-of-function and the ease of *Drosophila* genetics, this system represents an ideal model to perform screens for the identification of new genes involved in maintaining neuronal lipid homeostasis required for proper function of the photoreceptors *in vivo*, and this aspect is the main strength of this study. Importantly, the use of this system could also shed light on the mechanisms behind human neurodegenerative disorders, as many of these involve dysregulation of lipid signaling and lipid transfer at membrane contact sites. A novel and interesting finding is the identification of another lipid transfer protein, Cert, to be involved in the degeneration of photoreceptors.

The main limitation lies in the experimental design proposed by the authors to define the genes that are studied in their system. These are identified as potential interactors of human VAP in a mammalian cell line. Despite the fact that VAP is a highly conserved protein, and the genes identified are present in *Drosophila* as well, there is no evidence that these interactions are in fact occurring in *Drosophila* photoreceptors, and in fact, based on the function and the lack of VAP-binding motif in many of the 52 genes identified to have an effect on the RdgB phenotype, it is likely that many of the interactions are purely genetic and indirect, and that the modulation of the phenotype could in fact be due to a wide variety of factors (including, as discussed by the authors, gene expression, post-translational modifications, trafficking of proteins, etc) unrelated to mechanisms of VAP-RdgB mediated lipid transfer at ER-PM membrane contact sites. A more unbiased screen could have been carried out to identify VAP interactors involved in this degeneration phenotype by testing all of the FFAT or FFAT-related motif containing proteins. Due to this initial bias in the selection of genes to be tested, it is possible that other important VAP interactors that play a role at the ER-PM interface of photoreceptors have not been identified.

This study provides great functional advance in the understanding of genes implicated in photoreceptor degeneration, and in those regards it is a great resource for a specialized audience, as it enables further characterization by others of the different processes implicated in this neurodegeneration phenotype. However, the advance is small in regards to the core mechanism of RdgB function at VAP-mediated ER-PM, which was the main aim of the article and the most broadly interesting aspect of the study. Many of the VAP-interacting proteins identified in the proteomic approach were already expected to be VAP interactors as they contain FFAT motifs, and these FFAT-containing proteins do not seem to have a major role in VAP-RdgB maintenance of neuronal lipid homeostasis, with the exception of Cert. The implication of Cert in RdgB-mediated lipid homeostasis is certainly interesting as it touches on a current topic in the field of membrane contact sites related to how the multiple interactions of VAP, a universal contact site adaptor at the ER, are regulated and influenced by each other.

**Reviewer's field of expertise:* Lipid transfer at membrane contact sites; membrane lipid homeostasis in neurons. All of the *Drosophila* data seem to be of good general quality to me, but I do not have any expertise in *Drosophila* work.

December 15, 2023

Re: Life Science Alliance manuscript #LSA-2023-02525

Prof. Raghu Padinjat
National Centre for Biological Sciences
Cellular Organization and Signalling
TIFR GKVK Campus
Bangalore, Karnataka 560065
India

Dear Dr. Padinjat,

Thank you for submitting your manuscript entitled "A genetic screen to uncover molecular mechanisms underlying lipid transfer protein function at membrane contact sites and neurodegeneration" to Life Science Alliance. We invite you to re-submit the manuscript, revised according to your Revision Plan.

Thank you for this interesting contribution to Life Science Alliance. We are looking forward to receiving your revised manuscript.

Sincerely,

B. MANUSCRIPT ORGANIZATION AND FORMATTING:

Response to reviewers and revisions carried out on the manuscript Mishra et.al.

We thank the reviewers for their valuable comments. As indicated in the revision plan we have performed additional experiments and also revised the manuscript accordingly. In the manuscript file, changes made to the original version are highlighted in yellow. Responses to individual comments from reviewers are included below.

Reviewer #1 (Evidence, reproducibility and clarity (Required)):

Mishra and colleagues have conducted a large genetic screen to identify modulators of a Drosophila model for retinal degeneration. Using biomolecular techniques, they selected a few hundred proteins that interact with an ER bound protein VAP, to further test them in the retinal degeneration model. This was done by downregulating their expression using interference RNA (RNAi). Degeneration was first measured through a pseudopupil analysis, then suppressors of degeneration were further tested in a different retinal degeneration model, and finally through an ERG experiment. Finally, they focused on a strong suppressor of degeneration, dCert, by using a mutant allele to confirm the findings from the RNAi.

The results suggest that a handful of these candidates are suppressors in this model of retinal degeneration, and identify at which stage of retinal degeneration these proteins may be involved in. These proteins may have a significance in forms of neurodegeneration.

Major comment:

- Perhaps a larger number of replicates could be done in the optical neutralization experiment as well as in the ERG. Figure 4.A(i) and (ii), please clearly state n values. I would suggest this as optional, but perhaps it could help to increase n?

Each optical neutralization experiment was done using 5 independent animals and 10 ommatidia were scored from each animal. This captures the interommatidial variability in each eye and the inter-individual variability for each genotype; the number has been set based on our experience in doing these types of analyses for almost 20 years.

For the ERG analyses, a minimum of 5 animals were used per experiment ;this is already mentioned in the figure legend.

For the optical neutralization experiments, as mentioned above, in 4.A(i) and (ii) we did 5 animals with 10 ommatidia/animal for the statistical score. This information has been added in figure legends where it was not previously mentioned.

- For human orthologs (Table 1), it could be worthwhile to add alignment scores between fly and human?

In the revised Table 1, we have added a column depicting the extent of similarity between the human and fly homologs.

Minor comment:

- Clarify the purpose in focusing on dCert specifically in the last results section and discussion
- Several typos
- Affect vs effect

- Following the initial genetic screen, it was necessary to characterize a gene to understand in detail the temporal and spatial aspects of its role in modulating degeneration as well as verify the results obtained from the RNAi screen. Dcert was chosen for several reasons (i) a classical germ line mutant allele was available (ii) Prior papers had established its role as a protein that functions at contact sites. We will clarify our purpose of including dCert as proof of principle in the discussion part.

-Typos have been corrected.

Reviewer #1 (Significance (Required)):

General assessment:

The main significance of this study comes from the focus on proteins that are known to interact with VAP. This implies that the suppressors of degeneration that they have identified in the RdgB9 model may have an effect in other neurodegenerative models, namely in ALS models. This could have a very high significant potential in therapeutic avenues for neurodegenerative diseases.

Among six candidates that had an effect with knocked down through RNAi, they pursued a single one (dCert) as proof of principle. It would help to add a justification for this choice in the main text and whether the authors have performed or intend to perform experiments using mutant forms of the other candidate proteins.

Although six candidate genes were available for analysis, there were no mutants available in two of them (SET and CG3071). Mutants in Yeti are homozygous lethal making it difficult to work on it in this setting. A viable mutant in APC is available but this protein has no obvious membrane interaction domains. However, as dCert and CG9205 have membrane interacting domains we have focused on these two genes for this study as proof of principle. For CG9205 CRISPR germ line deletion mutant has recently been generated in our lab. We have used these germ line knockout alleles in dCert and CG9205 to test their interaction with rdgB. The results obtained for the detailed analysis of CG9205 are now included in the new Fig 5.

A small section on the rationale for choosing these mutants has been added in the results section.

The work from Raghu and his team have been leading the research surrounding this model of degeneration in Drosophila. This study naturally further extends their field of research, identifying more candidates that modulate this form of degeneration, and helping elucidate the pathways leading to cellular degeneration.

These results will be of high interest for specialized researchers studying the molecular pathways that lead to cellular degeneration, both in the context of retinal degeneration as well as neurodegeneration. Specifically, researchers that may be interested in these candidate proteins and how they may play a role in the pathogenesis of various degenerative diseases.

Reviewer #2 (Evidence, reproducibility and clarity (Required)):

Summary: In this study, Mishra and colleagues combine proteomic techniques with Drosophila genetics to identify interactors of the endoplasmic reticulum (ER) protein VAP and analyse their indirect implication in the light-dependent retinal degeneration phenotype caused by misfunction of the lipid transfer protein RdgB, another VAP interactor. This is an ideal model system to study neurodegeneration in vivo via which the authors aim to further understand the molecular mechanisms involved in VAP-RdgB's function in maintenance of membrane lipid homeostasis. Of an initial list of 403 VAP mammalian interactors found via Immunoprecipitation - Mass Spectrometry performed in a human cell line, 52 homologous Drosophila genes are found to suppress the RdgB degeneration phenotype upon knockdown. The authors then test a series of genetic interactions to dissect the potential role of these genes in the degeneration phenotype and identify six genes that likely suppress the phenotype by directly acting on the same molecular processes as RdgB, two of which (Cert and CG9205) could have a direct mechanistic role in the lipid homeostasis maintained by the VAP-RdgB complex.

Major comments:

- The model suggested by the authors in Fig 2A is one by which the tested genes influence the VAP-RdgB function via direct binding to VAP. The direct interaction with VAP is a core aspect of the study as the IP-MS experiment biases the list of genes tested in Drosophila, and this list is often referred to as "VAP interacting proteins". However, the interactions between the proteins coded by the most relevant genes in the genetic screen

and VAP are not tested. In fact, of the six genes that are found to likely modulate the same processes as RdgB, four probably do so via affecting gene expression, as discussed by the authors, therefore making it unlikely that they are true VAP interactors (unless they shuttle from the ER to the nucleus). Additionally, it would seem that most of the 52 genes found to suppress the degeneration phenotype are not necessarily VAP interactors either as only a handful of genes in this list have predicted strong FFAT motifs. In general, the authors should provide additional comments/evidence on the interaction, or likelihood of the interaction, of these proteins with VAP, that should include:

- o The IP-MS data should be made available (at least on my end, in the current submission the supplementary lists available cover only the 52 genes found to suppress the phenotype, I found the list of 403 genes in the biorXiv submission but this also does not include data on the enrichment of each hit in the IP-MS). It should be made clear how enriched were each of the proteins in the analysis, and how high the most relevant genes in the genetic screen rank within this interactor list.

We have now included the full mass spectrometry data:

Supplementary Table S1: The full list of VAP-A and VAP-B interacting proteins along with their FFAT and phospho-FFAT scores.

Supplementary Table S2: It contains the list of all proteins identified by MS/MS their score PSM number of peptides, FDR settings and search parameters for the MS analysis. This has also been explained in the results and the materials and methods.

- o The authors should provide further detail on the rationale behind defining the list of 403 genes to be tested (for example, what was the threshold enrichment considered as interaction). Also in relation to this, the authors should at least provide speculation as to why more than half of the 403 genes defined do not have an FFAT motif, despite the fact that the proteomic data was normalized to a non-FFAT motif binding mutant of VAP which should be capable of maintaining non-FFAT mediated interactions of the protein.

The overall purpose of our project was to identify molecules that might influence the function of RDGB. For this we started with the known observation that RDGB interacts with VAP through its FFAT motif (Yadav, *J.Cell.Sci* 2015). Therefore other proteins that interact with VAP and might be in the proximity might also interact with RDGB. To identify such proteins, we studied proteins identified as VAP interactors in a immunoprecipitation done with VAP identified using mass spectrometry.

It is important to highlight that in such a proteomics experiments all proteins that interact with VAP will be identified including two broad classes (i) Those that interact with VAP directly via the FFAT motif-such proteins would be those having an identifiable FFAT motif (ii) Those that do not have an FFAT motif and are therefore interacting with VAP indirectly, perhaps through another FFAT motif or through a non-FFAT mediated interaction with VAP. Both these classes of interactions are depicted in Fig 2A. In this proteomics study a total of 401 proteins encompassing both the above classes was identified.

This point has been discussed in detail in the revised version. We have also cited a previous study (Bioplex project) in which the VAP protein interaction network has been studied by mass spectrometry; this study has identified proteins such as LSG1 that interact with VAP that do not have a FFAT motif.

- o The authors should acknowledge the limitations of their experimental design in regards to identifying real interactors of VAP in Drosophila and avoid referring to this set of genes as "VAP interacting proteins" and rather use a more accurate description such as "proteins enriched in the IP-MS" or at least "potential VAP interacting proteins".

We have revised the terminology in our manuscript to reflect these limitations.

- o Testing interaction between VAP and all the 52 genes found to suppress the phenotype would be a huge amount of work. But the finding most relevant to the initial premise of the study (i.e. "molecular mechanisms

underlying lipid transfer protein function at membrane contact sites") is that Cert (a lipid transfer protein) and CG9205 (fly homolog of a mammalian lipid transfer protein) influence RdgB function. Demonstrating an interaction between these proteins and VAP would argue for the experimental design and support the hypothesis and model of the study. Cert is already a well established interactor of VAP, hence the authors would not need to add anything regarding this protein. Is CG9205 expected to be also a true interactor of VAP? Biochemical experiments could be used to test this idea, or even recently developed in silico modelling of interactions (i.e. AlphaFold Multimer) could be of help. If no interaction is observed/expected this should also be pointed out in the manuscript. Optionally, showing localization of Cert or CG9205 to the ER-PM interface would also greatly support the model of VAP-RdgB regulation suggested by the authors.

We have performed co-IP experiments between VAP and CG9205 using Drosophila head extracts (that includes eye tissue) to directly test for protein-protein interactions (Supp Fig S2). As expected we were able to demonstrate the well-established interaction between RDGB and VAP in our assay. We were also able to demonstrate the interaction between dCert and VAP, not previously shown in the fly but shown in other systems. Importantly, we were able to show an interaction between VAP and CG9205. These results have been considered in detail in the revised discussion.

Minor

comments:

- In the model shown in Fig2A, it would seem that many proteins can bind VAP in addition to RdgB, however, VAP proteins have only one FFAT binding pocket. This model would only be possible if oligomerization of VAP is considered (oligomerization of VAP has been reported to occur, see for example PMID:20207736). The model should be redrawn considering this fact.

It is important to remember that there are many molecules of VAP in a cell and not all of them need to interact with the same FFAT containing protein. Thus cellular VAP will likely consist of a collection of VAP molecular each of them engaged in an FFAT mediated interaction with a specific protein.

- In line 161 of the text VPS13D is mentioned, however VPS13C is the gene indicated in Fig 1D.

Corrected.

Reviewer #2 (Significance (Required)):

Previous studies by some of the authors and others have shown that RdgB can transfer lipids between the ER, to which it binds via VAP, and the plasma membrane (PM), and is required for proper replenishment of PI(4,5)P2 in the PM which is in turn necessary for sustained PLC signaling in Drosophila photoreceptors. Lack of RdgB leads to light-dependent degeneration of the retina, and hence it is utilized by the authors as a model for neurodegeneration. Given the clear phenotype of RdgB loss-of-function and the ease of Drosophila genetics, this system represents an ideal model to perform screens for the identification of new genes involved in maintaining neuronal lipid homeostasis required for proper function of the photoreceptors in vivo, and this aspect is the main strength of this study. Importantly, the use of this system could also shed light on the mechanisms behind human neurodegenerative disorders, as many of these involve dysregulation of lipid signaling and lipid transfer at membrane contact sites. A novel and interesting finding is the identification of another lipid transfer protein, Cert, to be involved in the degeneration of photoreceptors.

The main limitation lies in the experimental design proposed by the authors to define the genes that are studied in their system. These are identified as potential interactors of human VAP in a mammalian cell line. Despite the fact that VAP is a highly conserved protein, and the genes identified are present in Drosophila as well, there is no evidence that these interactions are in fact occurring in Drosophila photoreceptors, and in fact, based on the function and the lack of VAP-binding motif in many of the 52 genes identified to have an effect on the RdgB phenotype, it is likely that many of the interactions are purely genetic and indirect, and that the modulation of the phenotype could in fact be due to a wide variety of factors (including, as discussed by the authors, gene expression, post-translational modifications, trafficking of proteins, etc) unrelated to mechanisms of VAP-RdgB mediated lipid transfer at ER-PM membrane contact sites.

A more unbiased screen could have been carried out to identify VAP interactors involved in this degeneration phenotype by testing all of the FFAT or FFAT-related motif containing proteins. Due to this initial bias in the selection of genes to be tested, it is possible that other important VAP interactors that play a role at the ER-PM interface of photoreceptors have not been identified.

The starting point of this study was the identification of VAP interactors by mass spectrometry. An alternative approach would have been to identify, bioinformatically, all FFAT containing proteins encoded in the fly genome and perform the genetic screen only with them. The *Drosophila* genome has 52 protein coding genes with FFAT motifs (Raghu lab, unpublished data). Of these 12 were present in the list of VAP interactors identified by proteomics. It is possible that the rest were not expressed in the cells (human HeLa cells) from which the IP-MS was originally performed. It is possible that we may pick additional functional VAP interactors from the remaining 26 FFAT containing proteins in *Drosophila* through our genetic screen and may be seen as an alternative approach.

This study provides great functional advance in the understanding of genes implicated in photoreceptor degeneration, and in those regards it is a great resource for a specialized audience, as it enables further characterization by others of the different processes implicated in this neurodegeneration phenotype. However, the advance is small in regards to the core mechanism of RdgB function at VAP-mediated ER-PM, which was the main aim of the article and the most broadly interesting aspect of the study. Many of the VAP-interacting proteins identified in the proteomic approach were already expected to be VAP interactors as they contain FFAT motifs, and these FFAT-containing proteins do not seem to have a major role in VAP-RdgB maintenance of neuronal lipid homeostasis, with the exception of Cert. The implication of Cert in RdgB-mediated lipid homeostasis is certainly interesting as it touches on a current topic in the field of membrane contact sites related to how the multiple interactions of VAP, a universal contact site adaptor at the ER, are regulated and influenced by each other.

Reviewer's field of expertise: Lipid transfer at membrane contact sites; membrane lipid homeostasis in neurons. All of the *Drosophila* data seem to be of good general quality to me, but I do not have any expertise in *Drosophila* work.

February 22, 2024

RE: Life Science Alliance Manuscript #LSA-2023-02525R

Prof. Padinjat Raghu
National Centre for Biological Sciences
Cellular Organization and Signalling
TIFR GKVK Campus
Bangalore, Karnataka 560065
India

Dear Dr. Raghu,

Thank you for submitting your revised manuscript entitled "A genetic screen to uncover mechanisms of lipid transfer protein function at membrane contact sites". We would be happy to publish your paper in Life Science Alliance pending final revisions necessary to meet our formatting guidelines.

- please address Reviewer 2's remaining comments
- please be sure that the authorship listing and order is correct
- please upload your main and supplementary figures as single files
- please upload your Tables in editable .doc or excel format
- please add a Category for your manuscript in our system
- please add the Twitter handle of your host institute/organization as well as your own or/and one of the authors in our system
- please note that the titles in the system and on the manuscript file must match
- please add an Author Contributions section to your main manuscript text
- please add a conflict of interest statement to your main manuscript text
- please add callout for Tables S6 and S7 to your main manuscript text

Figure Checks:

- please add scale bars to microscopy images, indicating their size in the corresponding legend
- please add sizes next to all blots

A. FINAL FILES:

B. MANUSCRIPT ORGANIZATION AND FORMATTING:

Sincerely,

Reviewer #1 (Comments to the Authors (Required)):

Mishra and colleagues have conducted a large genetic screen to identify modulators of a Drosophila model for retinal degeneration. Using biomolecular techniques, they selected a few hundred proteins that interact with an ER bound protein VAP, to further test and confirm that several of them are highly likely suppressors in the retinal degeneration model. The results suggest that a handful of these candidates, in particular dCERT, suppress rdgB model of retinal degeneration, and identify at which stage of retinal degeneration these proteins may be involved in.

Reviewer #2 (Comments to the Authors (Required)):

The authors have provided the previously missing datasets and addressed the comments regarding the nature of the VAP interactions found in the discussion. Additionally, they have provided new data to deepen in the in vivo functional relationship between CG9205 and rdgB and to establish the main interactions of VAP studied in this paper as protein-protein interactions: with rdgB, dCert and CG9205. Both of these additions enrich the conclusions of the paper and increase the interest on the novel VAP interactor found, CG9205. I only have two comments:

- It is not clear why the negative control with a non-FFAT interactor for the co-IPs was not included (interactions with casein kinase is indicated as "data not shown" in the discussion).

- The explanation regarding Fig2A clarifies the previously made minor comment regarding the schematic. It was not clear before that the VAP molecule depicted in fact represents several VAP molecules, so in the schematic's current state it is somewhat confusing as it seems to suggest that one VAP molecule can bind to many targets at the same time, and that the removal of one binding partner will destabilize the VAP-RdgB interaction directly. Although this could be happening in some cases, the screen in this study considers also situations where VAP binding partners could be having more indirect effects by, for example, competing for the FFAT-binding pocket with RdgB, or stabilizing VAP at other ER-contact sites, or even in far more indirect ways. Perhaps including other VAP molecules would clarify this small confusion, or including an explanation of what "A", "B" or "C" could represent in the legends (e.g. interactor that directly stabilizes VAP-RdgB, FFAT containing competing interactor, non-

FFAT indirect interactor).

Reviewer #2 (Comments to the Authors (Required)):

The authors have provided the previously missing datasets and addressed the comments regarding the nature of the VAP interactions found in the discussion. Additionally, they have provided new data to deepen in the in vivo functional relationship between CG9205 and rdgB and to establish the main interactions of VAP studied in this paper as protein-protein interactions: with rdgB, dCert and CG9205. Both of these additions enrich the conclusions of the paper and increase the interest on the novel VAP interactor found, CG9205. I only have two comments:

- It is not clear why the negative control with a non-FFAT interactor for the co-IPs was not included (interactions with casein kinase is indicated as "data not shown" in the discussion).

This data has now been included in Supl Figure 2.

- The explanation regarding Fig2A clarifies the previously made minor comment regarding the schematic. It was not clear before that the VAP molecule depicted in fact represents several VAP molecules, so in the schematic's current state it is somewhat confusing as it seems to suggest that one VAP molecule can bind to many targets at the same time, and that the removal of one binding partner will destabilize the VAP-RdgB interaction directly. Although this could be happening in some cases, the screen in this study considers also situations where VAP binding partners could be having more indirect effects by, for example, competing for the FFAT-binding pocket with RdgB, or stabilizing VAP at other ER-contact sites, or even in far more indirect ways. Perhaps including other VAP molecules would clarify this small confusion, or including an explanation of what "A", "B" or "C" could represent in the legends (e.g. interactor that directly stabilizes VAP-RdgB, FFAT containing competing interactor, non-FFAT indirect interactor).

We have modified the legend of Figure 2A to explain the classes of functional interactors based on molecular mechanism.

March 5, 2024

RE: Life Science Alliance Manuscript #LSA-2023-02525RR

Prof. Padinjat Raghu
National Centre for Biological Sciences
Cellular Organization and Signalling
TIFR GKVK Campus
Bangalore, Karnataka 560065
India

Dear Dr. Raghu,

Thank you for submitting your Resource entitled "A genetic screen to uncover mechanisms of lipid transfer protein function at membrane contact sites". It is a pleasure to let you know that your manuscript is now accepted for publication in Life Science Alliance. Congratulations on this interesting work.

DISTRIBUTION OF MATERIALS:

Again, congratulations on a very nice paper. I hope you found the review process to be constructive and are pleased with how the manuscript was handled editorially. We look forward to future exciting submissions from your lab.

Sincerely,
